# Structural and Functional Diversity of Two ATP-Driven Plant Proton Pumps

**DOI:** 10.3390/ijms24054512

**Published:** 2023-02-24

**Authors:** Katarzyna Kabała, Małgorzata Janicka

**Affiliations:** Department of Plant Molecular Physiology, Faculty of Biological Sciences, University of Wrocław, Kanonia 6/8, 50-328 Wrocław, Poland

**Keywords:** plant proton pump, plasma membrane H^+^-ATPase, proton gradient, vacuolar H^+^-ATPase

## Abstract

Two ATP-dependent proton pumps function in plant cells. Plasma membrane H^+^-ATPase (PM H^+^-ATPase) transfers protons from the cytoplasm to the apoplast, while vacuolar H^+^-ATPase (V-ATPase), located in tonoplasts and other endomembranes, is responsible for proton pumping into the organelle lumen. Both enzymes belong to two different families of proteins and, therefore, differ significantly in their structure and mechanism of action. The plasma membrane H^+^-ATPase is a member of the P-ATPases that undergo conformational changes, associated with two distinct E1 and E2 states, and autophosphorylation during the catalytic cycle. The vacuolar H^+^-ATPase represents rotary enzymes functioning as a molecular motor. The plant V-ATPase consists of thirteen different subunits organized into two subcomplexes, the peripheral V_1_ and the membrane-embedded V_0_, in which the stator and rotor parts have been distinguished. In contrast, the plant plasma membrane proton pump is a functional single polypeptide chain. However, when the enzyme is active, it transforms into a large twelve-protein complex of six H^+^-ATPase molecules and six 14-3-3 proteins. Despite these differences, both proton pumps can be regulated by the same mechanisms (such as reversible phosphorylation) and, in some processes, such as cytosolic pH regulation, may act in a coordinated way.

## 1. Introduction

Three different types of proton pumps are present in plant cells, including the plasma membrane H^+^-ATPase (PM H^+^-ATPase), the vacuolar H^+^-ATPase (V-ATPase) and the vacuolar H^+^-PPase (V-PPase). In addition, plant cells possess two F-type ATPases that operate as ATP synthases in mitochondria and chloroplasts (Figure 1). The primary physiological function of proton pumps is the transfer of H^+^ across the membrane (plasma membrane or endomembrane), and consequently the generation of an electrochemical proton gradient [1]. For this reason, they play a key role in many physiological processes and cellular mechanisms activated in plants in response to unfavorable environmental conditions. These include the regulation of cytosolic pH [2], driving secondary active transporters involved in both the uptake and accumulation of nutrients and the removal of harmful substances [3,4] and cell elongation, in which the loosening of the cell wall and expansion of the central vacuole are essential [5,6,7,8]. Both PM H^+^-ATPase and V-ATPase, in contrast to V-PPase, use ATP as an energy source and ATP hydrolysis occurs as the first stage of enzyme functioning. However, despite the above-mentioned similarities, they vary not only in their localization within the cell, but, above all, in their structure. In general, PM H^+^-ATPase has been shown to be a single membrane-embedded polypeptide [4], while V-ATPase forms as a multi-subunit protein complex [9]. However, the use of modern techniques and tools allowed us to obtain the details of the structure of both enzymes and to identify the elements necessary for their functioning. It has been revealed that the PM proton pump at a state of high activity becomes a functional hexamer, composed of six ATPase molecules [10], while V-ATPase acts as a nanomotor, in which all subunits undergo conformational changes [11]. This review focuses on the current knowledge of the structure and related functional aspects of two plant H^+^-ATPases, highlighting the important differences between them.

## 2. PM H^+^-ATPase

The plasma membrane proton pump belongs to the P-type ATPases. This is a large family of proteins that pump ions across cell membranes. These enzymes form an aspartyl-phosphate intermediate throughout catalysis; therefore, they are termed P-type enzymes. P-type ATPases are divided into five subfamilies (P1A/B, P2A/B/C/D, P3A/B, P4, P5) [12], and the PM H^+^-ATPase belongs to the P3A subfamily. Plasma membrane proton pumps of this subfamily have been identified in plants and fungi. Plasma membrane H^+^-ATPase is an integral transmembrane protein. The principal role of plasma membrane H^+^-ATPase is to generate a proton gradient across the plasma membrane by pumping protons out of the cells [13]. This enzyme plays a central role in plant growth and development, as well as in the response to changing environmental conditions. Owing to the wide range of processes in which the plasma membrane proton pump is involved in various cells and tissues in plants, its regulation is complex and controlled at both the genetic and protein levels.

The structure of the functional protein in the plasma membrane proton pump is simple. It is a single polypeptide chain of approximately 100 kDa (Figure 2) [14]. Based on cryo-electron microscopic images of the PM H^+^-ATPase from fungi (*Neurospora crassa*) [15] and the crystal structure of the PM H^+^-ATPase isoform 2 (AHA2) from *Arabidopsis thaliana* [16], the plasma membrane proton pump has 10 transmembrane helices and three large cytoplasmic domains. Domains A (activator), M (membrane), P (phosphorylation), N (nucleotide binding) and R (regulatory) are distinguished in the protein. Three of these, P, N, and A, are responsible for ATP hydrolysis. The M domain is composed of 10 transmembrane helices. The PM H+-ATPase contains an autoinhibitor regulatory domain not only at the N-terminus but also at the C-terminus. Together, the N- and C-termini participate in the autoinhibition of enzyme proteins [17,18]. The details of the regulation of proton pump activity by its C- and N-termini are described below.

### 2.1. Multigene Family

In plants, the PM H^+^-ATPase is encoded by a gene family with a relatively large number of members [19]. Eleven H^+^-ATPase genes (*AHA1-11*) have been described in *Arabidopsis thaliana* [20], ten genes (*CsHA1-10*) in *Cucumis sativus* [21], twelve genes (*LHA1-10*) in *Lycopersicon esculentum* [22], ten genes (*OSA1-10*) in *Oryza sativa* [23], nine genes (*PMA1-9*) in *Nicotiana plumbaginifolia* [24] and four genes (*MHA1-4*) in *Zea mays* [25]. Members of this multigene family are grouped into five subfamilies [19]. *AHA4* and *AHA11*, together with *PMA1*, *PMA2*, *PMA3*, *OSA1*, *OSA2*, *OSA3*, *CsHA8*, *CsHA9* and *CsHA10*, belong to the first subfamily. *AHA 1*, *AHA2*, *AHA3* and *AHA5*, together with *PMA4*, *OSA7*, *CsHA1*, *CsHA2*, *CsHA3* and *CsHA4*, belong to subfamily II. *AHA 10*, *PMA9* and *OSA9* are clustered in subfamily III. *AHA6*, *AHA8* and *AHA9*, together with *PMA5*, *PMA6*, *OSA4*, *OSA6*, *OSA10*, *CsHA5*, *CsHA6* and *CsHA7*, belong to subfamily IV. *AHA7*, *PMA8* and *OSA8* are clustered in subfamily V [21]. The gene expression of some PM H^+^-ATPase subfamilies is not explicitly restricted to single organs, especially for members of the first and second subfamilies. The expression of genes encoding isoforms *AHA1*-*AHA4* and *AHA11* in *A. thaliana*, *PMA1*-*PMA4* in *N. plumbaginifolia* and *LHA1* in *L. esculentum* takes place throughout the plant, but mostly with different intensities in specific organs. In *A. thaliana*, *AHA1* is predominantly expressed in the shoots, while *AHA2* is in the roots. Similarly, in *Cucumis sativus*, the expression of genes belonging to the first subfamily (*CsHA8*-*CsHA10*) occurs evenly throughout the plant, in different tissues. The transcript of CsHA2-CsHA4 (encoded by genes belonging to the second subfamily) is also found throughout the plant, but CsHA2 and CsHA3 are more abundant in the roots and CsHA4 in the shoots.

The expression profile for genes belonging to the third, fourth and fifth subfamilies is different. These genes are only expressed in certain types of cells and tissues [19]. In *A. thaliana*, the genes *AHA6* and *AHA9* are expressed only in the anthers, and *AHA7* and AHA8 in the pollen [1]. In *C. sativus*, expression of *CsHA5-CsHA7* genes, included in subfamily IV, was only seen in the stamens and male perianth [21].

Interestingly, even in the same cell at the same stage of development, transcripts of different plasma membrane proton pump isoforms can be found [26]. In *N. plumbaginifolia*, the expression of two different PM H^+^-ATPase genes, *PMA2* and *PMA4*, has been found in guard cells [26]. This suggests that isoforms with distinct kinetics may coexist within the same cell. In contrast, in *Saccharomyces*, PM H^+^-ATPases are encoded by two genes, *PMA1* and *PMA2*. *PMA1* is expressed at high levels, whereas *PMA2* expression is very low [27].

### 2.2. Transport of Protons

The fundamental function of PM H^+^-ATPases as primary pumps is the unidirectional transport of protons. Several conserved amino acid residues play major roles in this process. PM H^+^-ATPase undergoes conformational changes during the catalytic cycle, giving rise to two distinct enzymatic states: Enzyme 1 (E1) and Enzyme 2 (E2). These two states, E1 and E2, are formed by the autocatalyzed formation and breakdown of the phosphoenzyme. This phenomenon is accompanied by the binding of protons on one side of the membrane and their translocation and release on the other side of the membrane. This is possible because, in the E1 conformation, the transmembrane binding site has a high affinity for the proton and ATP, while, in the E2, it has a low affinity for them [28].

Three domains (N, P and A) of the proton pump located in the cytoplasm play an essential role in ATP hydrolysis. During the catalytic cycle, these three cytoplasmic domains work together to autophosphorylate and dephosphorylate the enzyme. A special conserved Asp residue in the P domain is autophosphorylated. For the AHA2 isoform, this is Asp329. Phosphorylation is possible by binding ATP to the N domain. Later, dephosphorylation of the P domain is catalyzed by a Glu residue on the A domain. A conserved Asp residue exists in the membrane domain (M6). The formation of the E1P state of the enzyme is associated with the protonation of this amino acid residue. Conformational changes in M6 during the catalytic cycle contribute to the movement of this membrane part of the protein and, thus, to the transport of protons. During E1 phosphorylation, the proton-binding site is exposed on the cytoplasmic side. ATP-dependent phosphorylation results in an intermediate form of E1P that is spontaneously converted to E2P. This change shifts the proton-binding site outside the cytoplasm. E2P dephosphorylation contributes to the formation of E2, which spontaneously reverts to E1 [29].

The conserved Asp in the transmembrane segment M6 (in the *A. thaliana* isoform AHA2-Asp684) is a molecular factor involved in proton translocation, which functions as a proton acceptor/donor. The carboxylate side chain of Asp684 contributes to proton binding and pumping [28]. The Asn106 amino acid residue in the M2 domain, together with Asp684, plays an important role in controlling the efficient transport of protons against the electrochemical gradient. The arginine residue (Arg655) in the M5 domain, due to its location in the cavity opposite to Asp684, is considered to be a proton-return guard. Arg655 is in close contact with ASP684 (Figure 2). The positive charge of Arg655 allows the freedom of H^+^ from Asp684 and inhibits the attachment of extracellular protons to Asp684 [30].

Using the crystalline structure of PM H^+^-ATPase, AHA2 [16] and interactive molecular dynamic matching (iMDFF), scientists created an improved atomic model of AHA2 [31]. This more detailed model made some adjustments to the M domain. The improved structure of AHA2 allows for a more detailed structural and functional analysis of the proton transport mechanism. In particular, the protonation of Asp684 associated with phosphorylation (the E1P state) results from hydrogen bond interactions with Asn106. In the improved model, the minor cavity was visible above the Asn106-Asp684 pair. There are also two negatively charged amino acid residues, Glu113 and Glu 114 (M2), which can pull protons [31]. In the E2P state, a proton is released outside the cell membrane because of the formation of an internal salt bridge with the Arg655 residue [31]. Based on the modeling of E1P-E2P conformational changes, protons are likely transported via a large sol-vent-filled cavity that merges with an exit pathway toward the extracellular side of the membrane. Between segments M4 and M6, close to Asp684, towards the extracellular side, a large intramembrane cavity is visible. Most likely, protons are transported toward the extracellular side through this solvent-filled cavity. The presence of this cavity determines the expansion of the structure of the M4 helices on the conserved residues Pro286 and Pro290, as well as the bulged M6 structure at position Asp684. The Arg655 (M5) approaching the pair of Asp684 and Asn106 stimulates deprotonation [31].

It is assumed that PM H^+^-ATPase transports a single proton because of the hydrolysis of one ATP [20]. However, in some cases, the stoichiometry (H^+^/ATP) may be disturbed. There may be partial uncoupling between ATP hydrolysis and proton transport [29]. Potassium ions can induce the rapid dephosphorylation of the E1P enzyme, and, thus, K^+^ can act as a pump uncoupler [29]. K^+^ binds to Asp located in the P domain (Asp617 in AHA2). It has been suggested that potassium ions accelerate the docking of the A and P domains and promote the cleavage of the phosphoryl group located in the P domain. Potassium ions dephosphorylate the enzyme in E1P or E2P. If the E1P-E2P conversion occurs faster than the shift in domain A, E2P will be dephosphorylated by domain A and K^+^ will not be perceived as an uncoupler. On the other hand, if domain A moves before the E1P-E2P transition, K^+^ will act as a pump uncoupler [29].

### 2.3. Regulation by Phosphorylation and 14-3-3 Protein Binding

Activation of PM H^+^-ATPase results from the movement of the C-terminus of the enzyme due to the attachment of the regulatory protein 14-3-3 [32]. This is the most important regulatory mechanism for pump activation [33]. The carboxy terminus functions as an autoinhibition domain. This domain comprises the phosphorylation-site-specific penultimate threonine; in AHA2, this is Thr947 [34]. Phosphorylation of the penultimate Thr by appropriate kinases enables the attachment of the 14-3-3 protein. The 14-3-3 protein induces the movement of the C-terminal domain [14]. The binding of 14-3-3 proteins abolishes autoinhibition and activates pumping [35]. The phosphorylation of the penultimate Thr can be induced by several environmental factors [36]. The R domain probably inhibits PM H^+^-ATPase activity by physically blocking the rotation associated with the catalytic cycle [16].

On the one hand, proton pumps in land plants and fungi are well understood at the molecular level; on the other hand, knowledge about them in green algae is scarce [37]. It is already known that the C-terminal domain with the penultimate Thr as a crucial regulatory point is present only in terrestrial plant PM H^+^-ATPases. This means that this regulation comes from a later period of evolution. It has been shown that mosses and liverworts can have two types of PM H^+^-ATPases, with or without the penultimate threonine [38,39]. Pump regulation through the R domain is crucial for plant survival on land. The water-to-land transition of plants requires the precise control of growth and mineral nutrition [40]. Using AHA2 mutants with several truncated C-terminal variants, the importance of the C-terminal R domain in land plants has been demonstrated [40]. Mutated plants became more intensive in transporting protons out of the cell, which contributed to more intensive growth and increased nutrient uptake. Moreover, the change in the C-terminus contributed to the opening of the stomatal pores, which was disadvantageous because it allowed excessive water loss and the entry of pests. This means that the mutated plants were more sensitive to adverse environmental conditions, such as pathogen invasion or water deficits. Based on this study, the authors believe that the R domain is crucial for terrestrial plant fitness. The regulatory domain of plant PM H^+^-ATPase has evolved in some streptophyte algae, with a tendency toward terrestrialization. Streptophyte algae are the closest relatives of land plants [40].

Phosphorylation and dephosphorylation are important mechanisms that regulate the activity of many enzymatic proteins. The plasma membrane proton pump is regulated by its phosphorylation status. To date, many sites on this protein have been documented to undergo phosphorylation, consequently leading to an increase or decrease in its activity. [33]. The C-terminal penultimate Thr (Thr 947 in AHA2) was the first phosphosite identified in the plasma membrane H^+^-ATPase [41]. Eight additional phosphorylation sites have been identified in the C-terminal region of AHAs [32,42,43]. The contact of the 14-3-3 protein with the C-terminus of the proton pump covers a 28-amino-acid region. Within this region, various sites can be phosphorylated. Phosphorylation of sites other than the penultimate threonine may affect the binding of the 14-3-3 protein. Phosphorylation of Ser931 in AHA2 prevents 14-3-3 binding and lowers enzyme activity [44]. In *Nicotiana plumbaginifolia*, two phosphorylation sites (other than penultimate Thr, Thr955), Thr931 and Ser938, were identified [42]. Their phosphorylation interferes with the binding of 14-3-3 proteins, and, therefore, proton pump activation. Mutations in Thr931 or Ser938, although Thr955 is phosphorylated, are due to the absence of 14-3-3 protein binding to the C-terminus [42]. In plants, the plasma membrane pump is regulated by both N- and C-termini. Both regions participate in autoinhibition [18]. The N-terminal conformational change is coupled with changes that occur at the C-terminal end of the protein. Mutations in the N-terminal region of this protein have been shown to promote pump activity and yeast growth. Modification (mutation or removal) of the N-terminus results in the unmasking of the C-terminus, which allows protein kinases to phosphorylate the penultimate threonine and leads to the subsequent activation of the PM H^+^-ATPase [18].

### 2.4. Oligomerization

PM H^+^-ATPases have simple structures. They are composed of a single chain that crosses the membrane several times. However, they can also function as homo-oligomers. As previously mentioned, plant PM H^+^ ATPases can be activated by the phosphorylation of the penultimate Thr residue. This phosphorylation allows the binding of 14-3-3 regulatory proteins. Since 14-3-3 proteins form dimers with two binding sites [45], a 14-3-3 dimer likely binds two H^+^-ATPase molecules together, resulting in oligomerization. H^+^-ATPase exists as a dimer in the native plasma membrane of red beet. Similarly, cryo-electron analysis showed that the yeast-expressed PM H^+^-ATPase from *A. thaliana* functioned as a dimer [46,47]. Using native blue gel electrophoresis and chemical cross-linking, Kanczewska et al. [10] showed that the purified PMA2 isoform of *Nicotiana plumbaginifolia* was mainly unphosphorylated, free of 14-3-3 and, more importantly, present as a dimer. This proton pump is phosphorylated when expressed in yeast, and forms a complex with 14-3-3 proteins. In addition, it was shown that PMA2-14-3-3 had a circular structure with six symmetrical repetitions. This indicated that the complex contained six PM H^+^-ATPase molecules and six 14-3-3 proteins. On this basis, it is believed that the activation of the proton pump converts the dimer to a hexamer [10].

Three-dimensional reconstruction of the PM H^+^-ATPase/14-3-3 complex suggested a hexameric arrangement. Ottoman et al. [48] built a model of a holo complex based on the atomic structures of 14-3-3 and H^+^-ATPases. The model shows the location of 14-3-3 proteins in the upper part of the hexamer. In addition, the mass spectrometric cross-linking analysis showed that the R domain (C-terminus of the H^+^-ATPase protein) is involved in contacts between subunits in the hexamer (Figure 3) [49].

The hexamer architecture of fungal plasma membrane H^+^-ATPase has also been reported [14,50,51]. Cryo-electron microscopy analysis revealed the presence of 57 lipid molecules in the central hole of the PMA1 hexamer [51]. Similarly, a study on the structure of hexameric Pma1 from *Neurospora crassa* showed the important role of lipids in hexamers. Lipid-mediated contacts between monomers can act as proton-attracting funnels [52]. The importance of the role of lipids in the activity of the proton pump was also demonstrated in the PM proton pump from Pisum sativum reconstituted in artificial or native membranes. Sterol depletion contributed to the significant inhibition of active proton transport by the enzyme and an increase in passive H^+^ leakage [53].

Finally, it seems that the consideration of the simplicity of the structure of PM H^+^-ATPase, as a simple protein composed of only a single polypeptide chain anchored many times in the membrane, should be revised. In fact, when the proton pump is active, it transforms into a large twelve-protein complex of six H^+^-ATPase molecules and six 14-3-3 proteins.

## 3. V-ATPase

V-ATPases are phylogenetically the oldest and most complex proton pumps [1]. They share many features with F-type ATPases (ATP synthases) found in mitochondria and chloroplasts, as well as with archaebacterial A-ATPases, which suggests that these proteins have a common origin, although they differ in function [54]. Unlike PM H^+^-ATPase, V-ATPases are found in all eukaryotes, including human. In animal cells, V-ATPases function in the membranes of many organelles characterized by an acidic lumen, such as endosomes, lysosomes, the Golgi apparatus and clathrin vesicles. In neurons, they energize transporters located in the membranes of synaptic vesicles that are responsible for loading the vesicles with neurotransmitters [55]. Moreover, some specialized mammalian cells, e.g., intercalated cells of the kidney, osteoclasts or cancer cells, possess an additional subset of V-ATPases, present in the plasma membrane, which play a cell-specific function [56,57].

Knowledge about plant V-ATPases and their structure has evolved significantly since the 1980s, when vanadate-resistant and anion-sensitive ATP-dependent proton pumping activity was first demonstrated in the microsomal vesicles and isolated vacuoles [54]. For a long time, research on the functioning of V-ATPases in plant cells focused on their role in generating the proton gradient and driving secondary transport processes across the tonoplast. More recent studies showed that these proteins are present in other endomembranes and involved in vesicular transport in the secretory system [54]. It was confirmed that V-ATPases acidify not only the vacuoles but also the lumen of Golgi, the *trans*-Golgi network (TGN) and endosomes [9,58].

### 3.1. Overall Structure

Plant V-ATPase is a large complex of thirteen different subunits, with a total molecular weight of approximately 800 kDa, organized into two sectors: the catalytic peripheral V_1_ responsible for ATP hydrolysis and the membrane-integral V_0_ participating in proton translocation. Each of the two sectors forms a subcomplex composed of several subunits. Eight subunits, named VHA-A, B, C, D, E, F, G, H, and five subunits, including VHA-a, c, c″, d and e, were distinguished in V_1_ and V_0_, respectively [7,59,60]. It was shown in yeast and mammals that the V_1_ and V_0_ sectors of V-ATPase can dissociate and reassociate (reversible dissociation) in vivo in response to various stimuli, such as growth factors or changes in glucose concentration, thereby regulating V-ATPase activity. However, a similar mechanism has not yet been confirmed in autotrophic plant tissues [9]. The structures of the holoenzyme and its separated V_1_ and V_0_ sectors, as well as individual subunits and subunit subcomplexes, have been studied extensively, enabling the identification and characterization of interactions between individual subunits and detailed analysis of secondary structure elements [61].

#### 3.1.1. V_1_ Subcomplex

The first studies of V-ATPase using electron microscopy (EM) showed that the V_1_ sector has a “head and stalk” structure [62]. However, over the past two decades, significant progress has been made in learning about the V-ATPase complex and understanding the role of the individual subunits. The enzyme “head” is formed by three VHA-A (69 kDa) and three VHA-B (54 kDa) subunits, arranged in an alternating A_3_B_3_ hexamer (Figure 4A), with three ATP-binding domains (catalytic sites) located at three of the six AB interfaces [60]. Although both A and B subunits are characterized by the presence of nucleotide-binding sites, ATP hydrolysis takes place in VHA-A, while VHA-B lost this ability during evolution [7,9]. The hexamer resembles the catalytic head of ATP synthases. VHA-A, which dominates in the V_1_ sector, is composed of four domains (I–IV). Among them, domains I, III and IV are homologous to subunit β of ATP synthases, while domain II is specific to V-ATPases. Two nucleotide-binding P-loops were identified in domains III and IV, but only the first is involved in the hydrolase activity of the enzyme. The second P-loop was suggested to participate in the coupling of the catalytic head with the central stalk. VHA-B is a homolog of the α subunit of ATP synthases and has a regulatory function [9,60].

The remaining V_1_ subunits are organized into two types of “stalks”, one central stalk and three peripheral stalks, which connect V_1_ with V_0_ (Figure 4A). A heterodimer of single VHA-D (29 kDa) and VHA-F (16 kDa) subunits forms the central stalk, which fills the central pocket formed by a catalytic hexamer on the one side and is associated with the membrane subunits via VHA-d of V_0_ on the other side. VHA-F consists of two domains with a flexible C-terminus binding to the C-terminal part of VHA-B [9,60]. Three heterodimers of single VHA-E (25 kDa) and VHA-G (12 kDa) subunits (EG1, EG2 and EG3) function as peripheral stalks. Each EG heterodimer forms a long structure. It was shown in yeast that the E and G subunits interact more strongly at the N and C ends and weakly in the middle of the helices [63]. The globular C-terminal domains of EG stalks are in contact with the N-terminal parts of the B subunits in the catalytic hexamer. On the other hand, each of the EG N-termini, folded in an unusual right-handed coiled-coil, binds differently to VHA-C (43 kDa) and VHA-H (50 kDa), as well as VHA-a of the V_0_ sector. As a result, the only contact between the peripheral stalks and the membrane-embedded V_0_ is the interaction of EG2 with VHA-a [60,61,63]. In summary, the V_1_ subcomplex shows the stoichiometry of A_3_B_3_CDE_3_FG_3_H [57,64].

The A-G subunits are necessary for the assembly of the V-ATPase complex. In contrast, VHA-H is the only subunit that is not involved in this process [65]. It was shown that this subunit plays a dual role in enzyme complexes. Studies of yeast V-ATPase have confirmed that the H subunit is crucial for MgATPase activity and acts as an activator of the fully assembled enzyme. The complex lacking this subunit is inactive [65,66]. On the other hand, VHA-H is involved in silencing MgATPase activity in free V_1_ dissociated from membrane-embedded V_o_. It stabilizes one of the catalytic sites in the open state, resulting in the tight binding of inhibitory ADP at another site [67]. VHA-H consists of two globular domains, a larger N-terminal (H_NT_) and a smaller C-terminal (H_CT_), connected by a linker region [67]. In the V_1_V_o_ complex, H_NT_ interacts with one of the EG heterodimers (EG1), while H_CT_ binds to the a subunit (its N-terminal part, a_NT_); see Figure 4A. In autoinhibited V_1_, the interaction between H_NT_ and EG1 is maintained, but H_CT_ binds to the bottom of the A_3_B_3_ hexamer and the D subunit of the central stalk to prevent the rotation of the V_1_ sector [66,68]. Analysis of citrus V-ATPase by cryo-EM revealed that subunit H adopts two different conformations in the intact V_0_V_1_ complex, suggesting its additional role in enzyme regulation. Besides the one confirmation previously described, after rotation of the C-terminal domain (H_CT_), the H subunit binds to the AB dimer, interposing between the two subunits. This resembles the state observed in the free V_1_ subcomplex. For this reason, it has been proposed that this subunit may inhibit intact V-ATPase similarly to dissociated V_1_ [69]. Genome-wide analysis of VHA-H from different crop plants showed that the H_CT_ domain is shorter and more conserved than the H_NT_ [65].

VHA-C, similar to VHA-H, is composed of two globular domains, named as C_head_ and C_foot_, connected by a flexible linker. The head domain of VHA-C binds to EG3 with high affinity (C_head_-EG3, binary interface), while its foot domain interacts relatively weakly with both EG2 and the N-terminus of the a subunit (EG2-a_NT_-C_foot_, ternary interface); see Figure 4A [64,66]. It was found that the EG-C interaction is essential to maintain the stability of the EG heterodimer [63]. In contrast to VHA-H, which remains associated with V_1_ after V_0_V_1_ disassembly, the C subunit is released into the cytoplasm [61].

#### 3.1.2. V_0_ Subcomplex

The proteolipid ring consisting of VHA-c (16 kDa) and VHA-c″ (18 kDa) is the central element of the V_0_ sector in plant V-ATPases; see Figure 4A [60]. Until recently, six subunits were believed to be built in this part of the enzyme. However, the latest studies using yeast and mammalian cells have shown that ten subunits are involved in forming the proteolipid ring [55,70,71]. In contrast to plants and mammals, an additional proteolipid subunit, VHA-c′, is present in yeast V-ATPase [59]. Thus, it was demonstrated that the c-ring of yeast V-ATPase contains VHA-c, VHA-c′ and VHA-c″ in the ratio of 8:1:1 [70,71], whereas, in the mammalian brain V-ATPase, it includes nine copies of subunit *c* and one of subunit c″ [55]. Studies of the tonoplast proteome of *Arabidopsis* suggested that in plants, the c″ subunit is present in the ER and Golgi, but absent in the vacuole [72]. In 2022, using a purified V-ATPase from citrus fruit, it was confirmed that, as in mammals, the c-ring of the plant enzyme is composed of nine VHA-c and one VHA-c″ [69].

Both VHA-c and VHA-c″ are composed of four transmembrane α-helices [72]. A total of 40 helixes, derived from the 10 subunits, are arranged in two rings, an inner and an outer, with two helices from each proteolipid in each ring; see Figure 4B [70]. The proteolipid subunits have one lipid-exposed conserved glutamate residue, negatively charged, on their outer α-helices, which is responsible for proton transport during the catalytic cycle. In the VHA-c and VHA-c″ subunits, they are located in the fourth (TM4) and second (TM2) helices, respectively [9,60]. This arrangement results in an asymmetric distribution of Glu residues around the ring [69]. As was suggested for the yeast holoenzyme, VHA-c″ functions as the main molecular contact between the c-ring and the VHA-d subunit mediated by the N-terminus of the d subunit and the cytosolic loops of the c″ subunit [70,71].

It was demonstrated that the proteolipid ring of plant V-ATPase differs from that of yeast and mammals. Two additional transmembrane α-helices were found in the middle part of the c-ring of citrus, one related to the AP1 (accessory protein 1) and the other corresponding to the AP2 (accessory protein 2); see Figure 4B [69]. AP1 is homologous to the yeast Voa1p and partly to the mammalian Ac45 (ATP6AP1) that, similar to AP1, forms one of the middle helices. AP2, on the other hand, is a homolog of (pro)renin receptor PRR (ATP6AP2), present in the mammalian V-ATPase and absent in the yeast enzyme, suggesting that its presence is a feature of V-ATPases from higher eukaryotes. However, unlike in plants, in both mammals and *S. cerevisiae*, one of the additional transmembrane α-helices is derived from the c″ subunit. This gives a total of two or three helices within the center of the c-ring, respectively [69].

The VHA-d (40 kDa) is the only V_0_ subunit lacking transmembrane domains, located on the cytoplasmic side of the c-ring (Figure 4A). Such location makes it possible to block the central pore formed by the proteolipid ring [9,60,71]. It was shown in bovine brain V-ATPase that the d subunit forms several connections with c_1_, c_7_, c_8_, c_9_ and c″. Moreover, the results indicated that this subunit is a key element coupling the DF heterodimer with the c-ring [68]. In mung bean, the V_0_ sector was found to function as a passive proton channel in the absence of VHA-d. Consequently, this subunit was identified as a part of the V-ATPase central stalk [73].

The largest subunit of the V_0_, involved together with the proteolipid ring in proton pumping, is VHA-a (95 kDa), consisting of a C-terminal membrane-embedded domain (a_CT_) and an N-terminal hydrophilic domain (a_NT_) exposed to the cytosol (Figure 4A). The a_NT_ domain participates in linking V_1_ to V_0_. It is folded as a hairpin with two globular segments. In free V_0_, it interacts with subunit d, connecting subunit a to the c-ring-d subcomplex [71,74]. The C-terminal part (a_CT_) forms two offset hemi-channels that create a pathway for proton transport, one for proton entry from the cytoplasmic side of the membrane and the other for subsequent proton release into the vacuolar (organelle) lumen [60]. It includes eight transmembrane α-helices (TM1-TM8), of which the seventh (TM7) contains a positively charged arginine residue essential for proton translocation. Based on the side-directed mutagenesis of buried polar and charged residues in the a_CT_ of yeast V-ATPase, a working model of the proton-conducting hemi-channels was proposed. It assumes that the cytoplasmic hemi-channel is located at the interface of the c-ring with TM7/TM8 of the a subunit. On the other hand, the luminal hemi-channel is formed by TM3, TM4 and TM7 of this subunit [75].

The smallest V-ATPase subunit, VHA-e (8 kDa), is formed by two transmembrane α-helices. It is probably associated with the a subunit. Its function remains questionable [56,60]. In *Arabidopsis* cells, the presence of VHA-e was detected in ER and TGN, but not in the vacuolar membrane. Thus, it has been suggested that it may be involved in endomembrane-specific assembly or targeting of the V-ATPase complex [72]. Recently, the e subunit has been identified as a component of the V-ATPase structure purified from the endomembranes of citrus fruit [69]. In animal cells, VHA-e was shown to be essential for V-ATPase activity [56,64].

Additionally, in yeast, the membrane-embedded subunit f (YPR170W-B) associated with V_0_ was identified. Its function is also unknown, but it is probably involved in anchoring the static part of the complex in the membrane [70]. In mammals, RNAseK, a conserved metazoan protein, has been found as a homolog of the *f* subunit of *S. cerevisiae* [55]. Thus far, the f subunit has not been confirmed in plants. The citrus V-ATPase lacks this element [69]. Thus, in conclusion, the plant V-ATPase structure shares many features with yeast and mammalian enzymes, but it is distinguished by the absence of the f subunit and the unusual arrangement of the c-ring [69].

#### 3.1.3. Interactions with Other Proteins and Regulation at Post-Translational Level

Additional polypeptides have been shown to be involved in the function and regulation of plant V-ATPase activity, confirming that the V-ATPase complex is not limited to VHA subunits [60]. As mentioned earlier, VHA-B acts as a regulatory subunit that can interact with other cellular components through its N-terminal domain. In *Mesembryanthemum crystallinum*, the association of the B subunit with the glycolytic enzyme aldolase and enolase has been demonstrated. Moreover, aldolase has been shown to stimulate V-ATPase activity by increasing ATP affinity. It was confirmed that the interaction of glycolytic enzymes with V-ATPase subunits is important for salinity tolerance [76]. *Arabidopsis* VHA-B subunits have been found to bind to and stabilize F-actin. They are involved in the regulation of actin polymerization/depolymerization processes, playing an essential role in the remodeling of the actin cytoskeleton [77]. Similar interrelations between the B subunit and aldolase as well as microfilaments have been demonstrated in animals [56]. In tobacco guard cells, the interaction between phosphoinositide 3-kinase type I (PI3K) and the vacuolar VHA-B2 isoform has been confirmed. It promotes V-ATPase activation, vacuolar acidification and stomatal closure during leaf senescence [78]. In addition, at least two of the three VHA-B isoforms of *Arabidopsis*, V-ATPase, VHA-B1 and VHA-B2, directly bind to the protein kinase SOS2, an essential element of the salt overly sensitive (SOS) pathway, crucial for plant salinity tolerance [79]. On the other hand, the B subunit can act independently of the V-ATPase complex. The Arabidopsis VHA-B1 isoform has been discovered as a nuclear-specific partner for hexokinase 1, suggesting that it is involved in glucose signaling [80].

Other enzyme subunits, in addition to VHA-B, can also interact with additional elements. *Arabidopsis* VHA-C binds to and is phosphorylated by the WNK8 (with no lysine (K) 8) protein kinase [81], while barley VHA-A interacts with the 14-3-3 regulatory protein [82]. More recent studies have shown the interaction of the citrus VHA-c4 subunit with the ethylene response transcription factor CitERF13. The ERF-VHA interaction seems to be involved in citric acid accumulation [83]. In turn, in animals, the c subunit and the a2 isoform of V-ATPase bind to Arf (ADP-ribosylation factor), belonging to the Ras-superfamily of small GTPases, and its activator ARNO (cytohesin-2), respectively. Since the ability of the a2 subunit to interact with ARNO depends on the luminal pH, this subunit acts as a putative endosomal pH sensor [84,85].

Interactions between VHA subunits and other proteins indicate not only the involvement of V-ATPase in many cellular processes (such as glycolysis or signal transduction pathways), but also potential mechanisms regulating the activity of this enzyme. The biochemical regulation of V-ATPase includes the phosphorylation and binding of the 14-3-3 protein. Some types of protein kinases have been shown to phosphorylate VHA subunits in plants. These include the mentioned WNK8 and PI3K, but also CDPK1, which has been found to activate V-ATPase in barley. In addition, it was suggested that a C-terminal domain phosphatase-like2 (CPL2) mediates the dephosphorylation of VHA-C in *Arabidopsis* [7,8]. On the other hand, barley V-ATPase has been shown to be activated by blue-light-dependent phosphorylation and the interaction of VHA-A with the 14-3-3 protein [82]. Another important mechanism responsible for controlling V-ATPase activity is redox regulation, related to the post-translational modifications of conserved cysteine residues. The enzyme activity is diminished in the presence of oxidants. In *Arabidopsis*, Cys256 (corresponding to the bovine Cys254) of the VHA-A subunit has been suggested to participate in this process. In addition, two cysteine residues of VHA-E, Cys134 and Cys186, conserved in plants and involved in the formation of an intramolecular disulfide bond, seem to be responsible for the regulation of V-ATPase in plants [9].

### 3.2. Rotary Mechanism—Rotor and Stator Functions

V-ATPases are rotary nanomachines that use the energy released during ATP hydrolysis within the A_3_B_3_ hexamer to pump protons across the membrane. In their multi-subunit structure, consisting of approximately 30 polypeptides, both rotor and stator elements have been identified. Each of the AB heterodimers exposes a single catalytic site. Three AB heterodimers form a pseudo-symmetric trimer [11]. Using cryo-EM, it was demonstrated that after ATP hydrolysis, each of the three catalytic sites is in a distinct conformation, representing sequential structural changes. These are “open”, “tight” and “loose” states, characterized as no nucleotide, ATP binding and ADP and phosphate binding, respectively [85]. The “open” conformation creates a pocket open to the cytoplasm, with a high affinity for the ATP molecule [68].

Two motors can be distinguished in V-ATPases: one within the V_1_ sector, converting the chemical energy of ATP hydrolysis into mechanical energy (rotation of the central stalk), and the other in the V_0_ sector, transducing mechanical energy (rotation of the c-ring) into potential energy stored in the proton gradient [61]. Changes in the conformations between individual catalytic sites, driven by ATP, cause the rotation of the central stalk (DF dimer) and, then, via VHA-d, the rotation of the c-ring to transport protons across the membrane. The peripheral EG stalks, interacting with the C, H and a_NT_ subunits, function as stators. They form a rigid structure that prevents the co-rotation of the catalytic hexamer and keeps it stationary relative to the proton channel [9,11,74]. The proteolipid subunits rotate clockwise past the C-terminal transmembrane domain of the a subunit, a_CT_. Three rotational states of the c-ring relative to the a subunit were observed. In addition, long transmembrane α-helices of the a subunit interacting with the c-ring were shown [11,74]. In contrast to the three rotational states, observed in holo V-ATPase, the c-ring of the free yeast V_0_ was found to be in only one orientation, suggesting a unique ‘‘resting position’’ after V-ATPase dissociation [71].

Zhao et al. [11] proposed that the V_1_ sector with three different nucleotide-binding sites functions as a three-step motor, in contrast to the proteolipid ring with ten proton-binding sites operating as a ten-step motor. Thus, the hydrolysis of three ATP molecules (the rotation of the central stalk is 120° for each hydrolyzed ATP) promotes a complete rotation of the c-ring, transferring ten protons across the membrane and giving the 3ATP:10 H^+^ ratio. This results in a symmetry mismatch between V_1_ and V_0_ [11,57,61]. Zhao et al. [11] also found that almost all of the subunits forming both V-ATPase parts, the rotor and the stator, undergo coordinated conformational changes during three rotational states. These changes include, among others, the bending of the part of the D subunit in contact with the *d* subunit, wobbling of the d subunit relative to both the D and F subunits, pushing of the A and B subunits against the EG peripheral stalks, bending of the E and G subunits along their elongated region, twisting of the c subunits as well as swinging of the N-terminal domain of the a subunit parallel to the membrane [11].

During rotation, the glutamic acid residues of the proteolipid subunits, Glu137 in VHA-c and Glu108 in VHA-c″ (in yeast), transfer protons from the cytosolic half-channel to the luminal half-channel within the a_CT_. Both hemi-channels are filled with water and exposed charged and polar amino acid residues [57,75]. After protonation, each of the Glu residues can enter the lipid bilayer and release the proton after the c-ring rotates 360° [61]. A conserved Arg735 (in yeast) of the a_CT_, located at the interface of a_CT_ with the proteolipid ring, interacts with the glutamic acid residue via a salt bridge. It is involved in their subsequent deprotonation and proton transfer to the luminal hemi-channel [86]. Dysfunction of this residue uncouples the proton pumping activity from ATP hydrolysis [86]. The rotation breaks the stable salt bridge and brings the glutamic acid residue of the next c subunit closer [87].

Schep et al. [88] proposed a model of the proton translocation pathway in *S. cerevisiae*. It assumes that two amino acid residues of a_CT_, Glu721 and Ser728, participate in the formation of the cytoplasmic hemi-channel, which is at least partially accessible from the cytoplasm. A proton enters the half-channel and binds to Glu137 of the c subunit, activating c-ring rotation. As a consequence, another proton is released from the c-ring via the luminal hemi-channel. This half-channel is partly formed by Asp425, Asp481 and His743. The essential Arg735 is located between the two hemi-channels. Most of the identified amino acid residues are present in α-helices 7 and 8 of a_CT_, which are in contact with the c-ring [88]. Furthermore, Ser792 and His796 have been shown to interact with c-ring glutamates prior to their interaction with Arg735, allowing deprotonation near the luminal cavity [71]. On the other hand, two aromatic residues of subunit a, Tyr733 and Trp737, located close to Arg735, were identified as crucial elements of the catalytic cycle. It was proposed that they maintain the arginine residue in the hydrophobic environment [89].

Analysis of the structure of the yeast proton channel in the lipid nanodisc showed that the proton translocation from the c-ring (Glu residues) to the luminal hemi-channel of the a_CT_ involves transient water wires. The proton is transferred from glutamic acid residues to Glu789 in the a subunit via water molecules. This residue acts as a proton acceptor and gating element for alternating bulk water access [74]. Roh et al. [71] suggested that proton transfer from Glu137 to Glu789 takes place via H-bonding to Tyr66 in the proton binding site. From Glu789, the proton is transferred to the luminal hemi-channel through His743, Asp425, Asp481 and other residues [71].

### 3.3. Isoforms of VHA Subunits

The activities of V-ATPases located in various organelles are regulated at multiple levels, including gene expression. Many VHA subunits are expressed as different isoforms [90]. In yeast cells, V-ATPase subunits are encoded by single-copy genes (Table 1). The exception is the a subunit with two isoforms, Vph1p and Stv1p [59]. Similarly, in *Arabidopsis*, single-copy genes were identified for most of the subunits forming the V_1_ sector, including VHA-A, VHA-C, VHA-D, VHA-F and VHA-H. In contrast, all subunits representing the V_0_ sector, as well as VHA-B, VHA-E and VHA-G of the V_1_ subcomplex, are encoded by multigene families. Summarizing, 28 *VHA* genes, encoding 13 V-ATPase subunits, were found in the *Arabidopsis* genome [59]. Since then, it has been shown that plants vary significantly in the total number of VHA isoforms, ranging from 15 in *Chlamydomonas reinhartdii*, 20 in *Cucumis sativus* and *Citrus sinensis* and 24 in *Oryza sativa* to up to 48 in *Malus* × *domestica* and 54 in *Glycine max*; see Table 1 [54,91,92,93]. Of all the VHA polypeptides forming the V-ATPase complex in plants, the proteolipid c subunit is encoded by the largest gene family, which may include up to 10 members, identified in *Glycine max* [54]. However, at the protein level, some c subunit isoforms are very similar or identical, as confirmed for five Arabidopsis VHA-c and three cucumber VHA-c [9,91].

Changes in the isoform composition of V_1_V_0_ complexes may influence their catalytic properties and subcellular localization, as well as determining regulatory mechanisms [90]. Two isoforms of the yeast *a* subunit differ in subcellular localization. Vph1p and Stv1p have been found in the tonoplast and Golgi, respectively [59]. Studies of the yeast *a* subunit indicated that organelle targeting information is located in a_NT_. Moreover, the W_83_KY sequence within the a_NT_ of Stv1p was identified as a signal necessary for targeting the Golgi. This localization depends on the specific interaction between the Lys84 residue of the motif and phosphatidylinositol 4-phosphate, PI(4)P [87,94]. On the other hand, it has been suggested that Vph1p may interact with phosphatidylinositol 3,5-bisphosphate, PI(3,5)P2, binding directly or indirectly to the lipids in the vacuolar membrane [90].

Using purified V-ATPase complexes from yeast, it was shown that the enzyme containing Vph1p hydrolyzes ATP at a higher rate than that containing Stv1p. In addition, this difference is dependent on the presence of lipids [95]. Vasanthakumar et al. [95] found that there are differences in the electrostatic surface charges near the opening of the cytoplasmic hemi-channels in Vph1p and Stv1p. Vph1p exhibits a more negatively charged surface related to the presence of two acidic residues, Glu706 and Asp707, in contrast to Stv1p, with a more positively charged surface, exposing three basic residues, Arg606, Lys608 and Lys611. These charges may be responsible for the observed differences in the activity of both types of complexes [95].

In *Arabidopsis*, the a subunit is encoded by three genes. The VHA-a1 isoform has been identified in the V-ATPase complex found in TGN, whereas VHA-a2 and VHA-a3 are present in the tonoplast enzyme [60]. It was shown that the TGN targeting sequence is located within the first 228 aa of VHA-a1 [96]. Recently, Lupanga et al. [58] identified a region involved in both ER export and TGN targeting, described as VHA-a1 targeting domain (a1-TD). This motif is specific to seed plants and differs from the W_83_KY sequence [9,58].

In higher organisms, some isoforms may be selectively expressed in specific cell/tissue types with specialized roles; others are ubiquitous and function as part of the housekeeping V-ATPases [56]. Moreover, in animals, some cell/tissue types have been found to contain an enriched population of isoforms involved in specialized functions, but the same subpopulations may also coexist with V-ATPases composed of different isoforms in other compartments of the same cell [97]. One example is the a subunit, present in four isoforms in human (Table 1). The a1 and a2 isoforms are ubiquitous and located in the endomembranes of many cells. On the other hand, the a3 and a4 subunits are associated with V-ATPases targeted to the plasma membranes of osteoclasts/pancreatic beta cells and intercalated cells/proximal tubules of the kidney, respectively [56,87].

Tissue- or cell-dependent expression of different VHA isoforms has also been confirmed in plants. Different functional specialization has been demonstrated between the three Arabidopsis VHA-E isoforms, including VHA-E3, identified as the epidermis-specific isoform, and VHA-E2, as the pollen-specific isoform [98]. Similar to VHA-E2, the *Arabidopsis* VHA-G3 is expressed in pollen, suggesting that both peripheral stalk subunits show some specificity in the formation of EG heterodimers [60]. Divergent patterns of tissue localization were found for two isoforms of the A subunit, VHA-A1 and VHA-A2, in tomato. *VHA-A1* expression is ubiquitous in tomato tissues, including roots, leaves, stems, flowers and fruits, while the *VHA-A2* transcript has been detected in roots and mature fruits. In addition, the *VHA-A1* level significantly increases in leaves in response to salinity, in contrast to *VHA-A2*, whose expression remains unchanged [99]. Among the three VHA-c isoforms in cucumber, *VHA-c3* is expressed at a constant level in all old and young tissues. On the other hand, *VHA-c1* and *VHA-c2* are upregulated in roots under copper [91]. The presence of different isoforms suggests some flexibility in the creation of both V-ATPase sectors. It is assumed that, in addition to a diverse subcellular localization, different VHA isoforms may assemble into V_0_V_1_ complexes characterized by distinct properties depending on the current needs related to the developmental state or physiological condition [9,62].

## 4. Coordination of Plant Proton Pump Functions

Experimental data indicate that both proton pumps are regulated by similar mechanisms and can act in a coordinated way, participating in the same processes. Among others, these proteins are involved in the maintenance of cytosol pH and signal transduction pathways activated in response to environmental stress factors [2,8]. Reversible phosphorylation mediated by cytosolic kinases and 14-3-3 protein binding have been proposed to be regulatory events leading to the coordinated activation of all three plant proton pumps (PM H^+^-ATPase, V-ATPase and V-PPase). On the other hand, reactive oxygen species can act as negative regulators of these enzymes. It is believed that both mechanisms, responsible for controlling the activity of the proton pumps, are involved in maintaining the optimal pH of the cytosol and adjusting it to the actual needs. In addition, secondary transporters contribute to cytosolic pH regulation and their activity may also be controlled by phosphorylation [2].

Proton pumps participate in stomatal movements. PM H^+^-ATPase functions as a key player in blue-light-mediated stomatal opening. In *Arabidopsis* guard cells, the AHA1 isoform plays the main role in this process. As a result of blue light action, photoreceptors, phototropins PHOT1 and PHOT2, are activated by autophosphorylation. This induces a cascade of events leading to the phosphorylation of a penultimate Thr residue in AHA1 and its activation. Plasma membrane hyperpolarization and an electrochemical proton gradient drive K^+^ influx through potassium channels and active transporters, respectively [8,100]. Light also induces the circadian cycle of V-ATPase gene expression and activity (as well as enhancing the production of PPi used by V-PPase). The trans-tonoplast proton gradient is involved in the vacuolar acidification and accumulation of solutes in guard cells. It was shown in *Arabidopsis* that the vacuolar pH is more acidic during stomatal closure than during opening. In contrast to the PM H^+^-ATPase, the action of vacuolar proton pumps was found to be required for ABA-induced stomatal closure. Moreover, it has been suggested that proton pumping is also essential to maintain a steady tonoplast potential [100,101].

The relationship between V-ATPase and PM-ATPase has been shown to play an important role in *Arabidopsis* responses to oxidative stress. It was proposed that the vacuolar VHA-d2 subunit affects H^+^ flux through the regulation of *AHA* gene expression. Under oxidative stress, proton efflux in the roots of the *vha-d2* mutant may be due to higher *AHA1* or *AHA2* expression [102].

Both tonoplast and plasma membrane proton pumps are involved in the distribution of organic acids (OAs) into the vacuole or apoplast, which determines *Arabidopsis*’ tolerance to aluminum toxicity. It was suggested that, in response to Al stress, PM-ATPase and plasma membrane secondary transporters, responsible for the OA efflux from the cytosol to the outside of the cell, are activated. Under these conditions, V-ATPase is inhibited (expression of *VHA-a2* and *VHA-a3* is downregulated). When this OA distribution pathway is impaired, the *VHA-a2* and *VHA-a3* genes are reactivated and V-ATPase drives the transport of OA from the cytosol to the vacuole [103].

It is well known that the SOS pathway, in which the SOS2–SOS3 complex phosphorylates and activates SOS1 (plasma membrane Na^+^/H^+^ antiporter), is induced during salinity to remove sodium ions into the apoplastic space. The plasma membrane H^+^-ATPase is required to energize SOS1 transport activity. On the other hand, the SOS2 kinase interacts with V-ATPase and enhances its activity, which is needed for the functioning of the tonoplast NHX1 antiporter, responsible for Na^+^ excretion into the vacuole [79].

## 5. Conclusions and Future Prospects

One of the most interesting issues is that both proton pumps can function side by side in the same membrane. A subset of P_3A_-type ATPases was identified in the vacuolar membrane. These include PH5, found in the petals of petunia flowers, and AHA10, expressed in the endothelium of *Arabidopsis*’ seed coat. They function as tonoplast proton pumps (besides V-ATPase and V-PPase), responsible for vacuolar acidification and the generation of the proton motive force responsible for the transport of proanthocyanidin (PA) precursors into the vacuolar lumen [104]. It has been suggested that the transmembrane proton gradient generated by P_3A_-ATPase is involved in energizing the MATE-type antiporter that carries PA precursors. On the other hand, the vacuolar sequestration of anthocyanins may be driven by MATE transporters that are powered by other proton pumps, such as V-ATPase [105]. Recently, a gene of the P-type ATPase proton pump, *Ma10*, was identified in apple. It encodes a tonoplast-localized proton pump that plays an important role in fruit vacuolar acidification by regulating the accumulation of organic acids [106]. Tonoplast P_3A_-ATPases have been proposed to be essential for certain cell types with extremely acidic vacuoles. In such cells, the H^+^ pumping capacity of the V-ATPase appears to be insufficient to generate a highly acidic pH in the vacuoles. This is related to the specific physiological functions of the vacuoles in special plant tissues [8].

Since it has been confirmed that both ATPases can be present in the same membrane, it seems interesting to investigate whether they can interact with each other. Very recently, analysis of the *Arabidopsis* interactome gained some new insights into the interaction network of the plant V-ATPase. It has been shown that among 448 interactions, VHA subunits interact with other membrane transporters, including the plasma membrane aquaporins PIP1B and PIP2A, the ammonium transporter AMT1;3 and the phosphate transporter PHT3 [107].

## Figures and Tables

**Figure 1 ijms-24-04512-f001:**
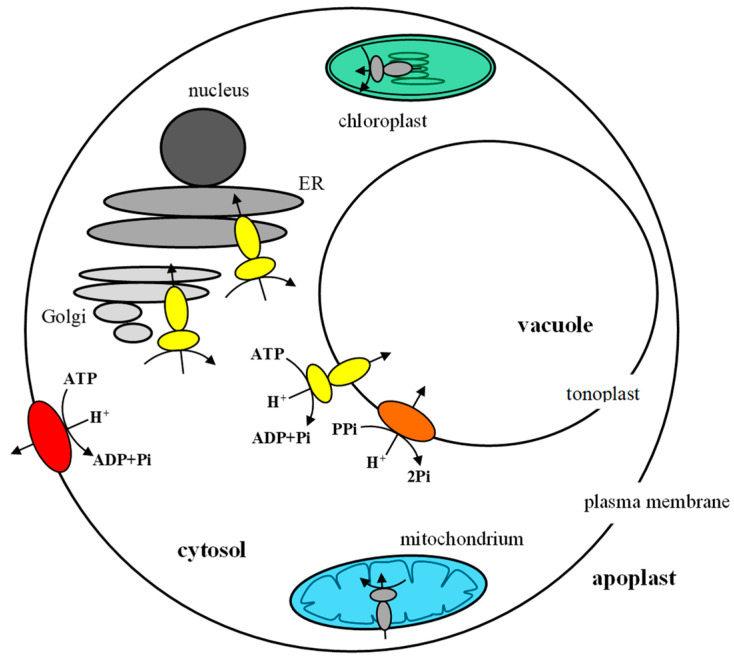
Proton pumps in plant cells. P-type H^+^-ATPase (red), located in the plasma membrane, pumps protons from the cytoplasm to the outside of the cell. V-ATPase (yellow) is present in the tonoplast and other endomembranes (ER, Golgi). At tonoplasts, it transports protons into the vacuolar lumen together with V-PPase (orange). Mitochondria and chloroplasts contain ATP synthase (F-type ATPase), responsible for ATP generation, coupled with H^+^ translocation from the intermembrane space to the mitochondrial matrix or from the thylakoid lumen to the stroma, respectively.

**Figure 2 ijms-24-04512-f002:**
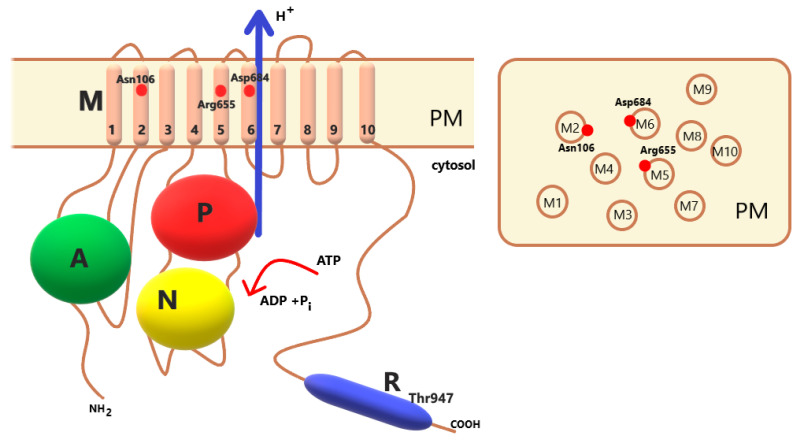
Schematic view of P-type ATPases. In the structure of PM H^+^-ATPase, we distinguish the A, M, P, N and R domains. Three cytoplasmic domains, phosphorylation (P, red), nucleotide binding (N, yellow) and actuator (A, green), are responsible for ATP hydrolysis. The M domain consists of the ten transmembrane helices. The C-terminal R domain contains the penultimate threonine (in AHA2 Thr947), which plays an important role in enzyme activation. On the right is a schematic view of the arrangement of the M1-M10 transmembrane domains in the plasma membrane. The red dots mark the amino acids that play a key role in the transport of protons (in AHA2, these are Asn106, Arg655 and Asp684).

**Figure 3 ijms-24-04512-f003:**
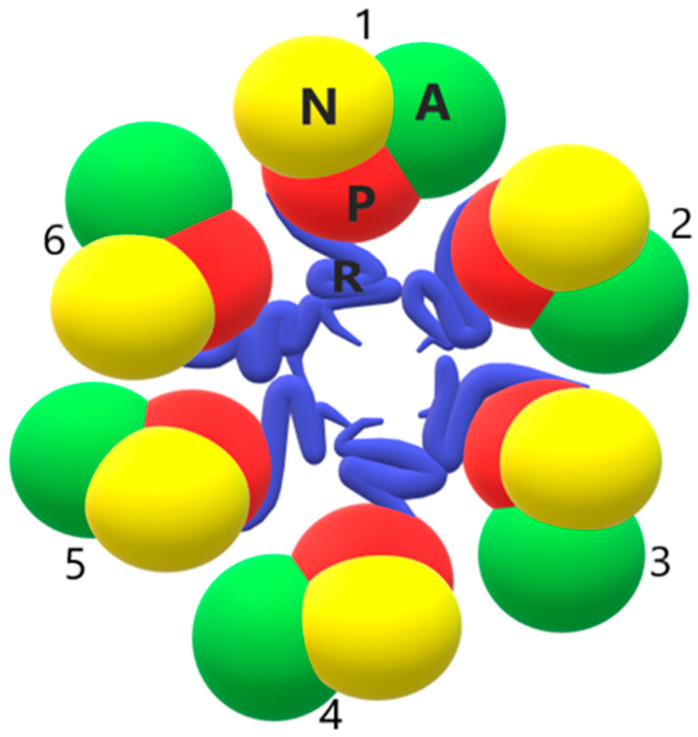
Schematic model of a functional PM H^+^-ATPase hexamer. The C-terminal regulatory domain mediates subunit–subunit contacts in the hexamer. The numbers 1–6 represent the individual PM H^+^-ATPase molecules. In each proton pump molecule, the N domain is shown in yellow, the P domain in red, the A domain in green and the R domain in blue. The M domain is omitted from the figure. The figure shows only the arrangement of the N, P and R domains of plasma membrane H^+^-ATPases during oligomerization.

**Figure 4 ijms-24-04512-f004:**
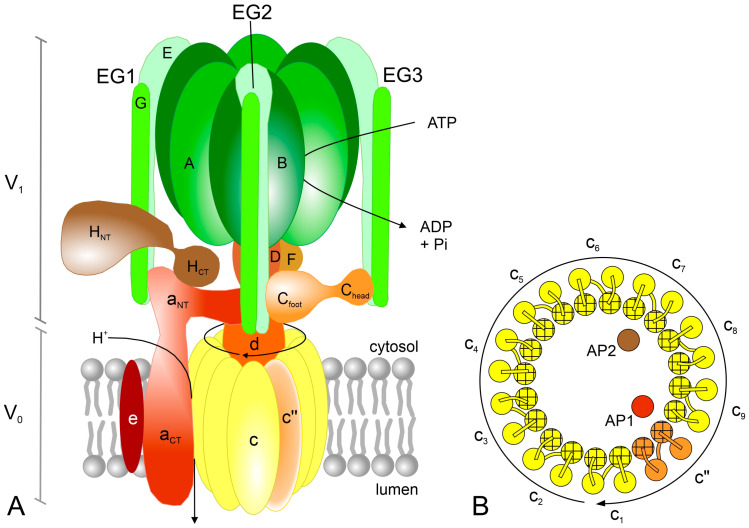
Structure of plant V-ATPase. V-ATPase is a complex of thirteen different subunits organized into two sectors, V_1_ and V_0_. (**A**) Overall structure. V_1_ sector is composed of subunits A, B, C, D, E, F, G and H. V_0_ sector consists of subunits a, c, c″, d and e. (**B**) Model of a proteolipid ring composed of nine c subunits and one c″ subunit; each subunit shows four transmembrane domains. AP1 and AP2 are additional transmembrane helices located in the middle of the c-ring.

**Table 1 ijms-24-04512-t001:** Isoforms of V-ATPase subunits in different eukaryotes.

Subunit	*Chlamydomonas* *reinhardtii*	*Citrus* *sinensis*	*Cucumis* *sativus*	*Oryza* *sativa*	*Arabidopsis* *thaliana*	*Malus* × *domestica*	*Glycine* *max*	*Homo* *sapiens*	*Saccharomyces* *cerevisiae*
V_1_ sector									
A	VHA-A	VHA-A	VHA-A	VHA-A1	VHA-A	VHA-A1	VHA-A1	ATP6V1A	Vma1p
				VHA-A2		VHA-A2	VHA-A2		
B	VHA-B	VHA-B	VHA-B	VHA-B1	VHA-B1	VHA-B1	VHA-B1	ATP6V1B1	Vma2p
				VHA-B2	VHA-B2	VHA-B2	VHA-B2	ATP6V1B2	
					VHA-B3		VHA-B3		
							VHA-B4		
C	VHA-C	VHA-C	VHA-C	VHA-C	VHA-C	VHA-C1	VHA-C1	ATP6V1C1	Vma5p
						VHA-C2	VHA-C2	ATP6V1C2	
						VHA-C3	VHA-C3		
						VHA-C4	VHA-C4		
D	VHA-D	VHA-D	VHA-D	VHA-D	VHA-D	VHA-D1	VHA-D1	ATP6V1D	Vma8p
						VHA-D2	VHA-D2		
E	VHA-E	VHA-E1	VHA-E	VHA-E1	VHA-E1	VHA-E1	VHA-E1	ATP6V1E1	Vma4p
		VHA-E2		VHA-E2	VHA-E2	VHA-E2	VHA-E2	ATP6V1E2	
				VHA-E3	VHA-E3	VHA-E3	VHA-E3		
						VHA-E4	VHA-E4		
							VHA-E5		
							VHA-E6		
							VHA-E7		
F	VHA-F	VHA-F1	VHA-F	VHA-F1	VHA-F	VHA-F1	VHA-F1	ATP6V1F	Vma7p
		VHA-F2		VHA-F2		VHA-F2	VHA-F2		
						VHA-F3			
G	VHA-G	VHA-G	VHA-G1	VHA-G1	VHA-G1	VHA-G1	VHA-G1	ATP6V1G1	Vma10p
			VHA-G2	VHA-G2	VHA-G2	VHA-G2	VHA-G2	ATP6V1G2	
					VHA-G3	VHA-G3	VHA-G3	ATP6V1G3	
						VHA-G4	VHA-G4		
						VHA-G5	VHA-G5		
						VHA-G6			
H	VHA-H	VHA-H1	VHA-H	VHA-H	VHA-H	VHA-H1	VHA-H1	ATP6V1H	Vma13p
		VHA-H2				VHA-H2	VHA-H2		
V_0_ sector									
a	VHA-a1	VHA-a1	VHA-a1	VHA-a1	VHA-a1	VHA-a1	VHA-a1	ATP6V0a1	Vph1p
	VHA-a2	VHA-a2	VHA-a2	VHA-a2	VHA-a2	VHA-a2	VHA-a2	ATP6V0a2	Stv1p
	VHA-a3		VHA-a3	VHA-a3	VHA-a3	VHA-a3	VHA-a3	ATP6V0a3	
						VHA-a4	VHA-a4	ATP6V0a4	
						VHA-a5	VHA-a5		
						VHA-a6	VHA-a6		
						VHA-a7	VHA-a7		
							VHA-a8		
c	VHA-c	VHA-c1	VHA-c1	VHA-c1	VHA-c1	VHA-c1	VHA-c1	ATP6V0c	Vma3p
		VHA-c2	VHA-c2	VHA-c2	VHA-c2	VHA-c2	VHA-c2		
		VHA-c3	VHA-c3	VHA-c3	VHA-c3	VHA-c3	VHA-c3		
		VHA-c4		VHA-c4	VHA-c4	VHA-c4	VHA-c4		
					VHA-c5	VHA-c5	VHA-c5		
						VHA-c6	VHA-c6		
						VHA-c7	VHA-c7		
						VHA-c8	VHA-c8		
						VHA-c9	VHA-c9		
							VHA-c10		
c′			-	-	-	-	-	-	Vma11p
c″	VHA-c″	VHA-c″	VHA-c″1	VHA-c″	VHA-c″1	VHA-c″	VHA-c″1	ATP6V0b	Vma16p
			VHA-c″2		VHA-c″2		VHA-c″2		
d	VHA-d	VHA-d	VHA-d	VHA-d	VHA-d1	VHA-d1	VHA-d1	ATP6V0d1	Vma6p
					VHA-d2	VHA-d2	VHA-d2	ATP6V0d2	
e	VHA-e	VHA-e	VHA-e1	VHA-e	VHA-e1	VHA-e1	VHA-e1	ATP6V0e	-
			VHA-e2		VHA-e2	VHA-e2	VHA-e2		
						VHA-e3	VHA-e3		
						VHA-e4	VHA-e4		
f	-	-	-		-	-	-	RNAseK	YPR170W-B
total	15	20	20	24	28	48	54	23	15

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
