# Peer review of "Structural and Functional Diversity of Two ATP-Driven Plant Proton Pumps"

_ijms, 2023, doi:10.3390/ijms24054512_

Round 1
Reviewer 1 Report
This review from Kabała and Janicka analyzes structural and functional diversity of two ATP-driven plant proton pumps. In this review the authors give a thorough and comprehensive overview of the literature concerning the structure of proton pumps and proton transport. A succinct style makes this work a very useful guide for researchers and students. Here are several comments on how to improve the quality and impact of this work.
1, Lacks clarity of thoughts and message throughout the text, the body of the review is not well organized to enable a reader to follow the message logically.
2, Although the manuscript is generally well written, it needs a careful English revision. Indeed, there are many English grammar mistakes throughout the paper, that need to be corrected.
3, It would be useful to list the functions of the different subunits of the two enzymes and phenotypes of subunit mutants in tables for the general readership.
4, The Summary and Perspective section reads more like a general summary. I recommend highlighting relevant knowledge gaps.
5, Refference style is not identical (i.e. abbreviation of Journal Name), it should be to keep identical in style of refference.
6, line 15, “thirteen different subunits”, should be “at least thirteen different subunits”.
Author Response
Reviewer 1
Thank you very much for all suggestions. We have tried to improve the manuscript according to all the points indicated.
- The abstract does not reflect the whole story, revise it.
The abstract has been corrected.
- The writing style of the paper is very poor. There are lots of grammatical mistakes. Long sentences with noticeable grammatical mistakes are frequently present throughout the manuscript. There are many typos mistakes in this whole manuscript. The author should check the whole manuscript.
We carefully checked the whole manuscript. Probably, some of the indicated errors are related to the use and notation of subunit abbreviations, e.g. the a subunit (i.e. subunit a of V0). To avoid such grammatical doubts, we have changed the notation of all subunits of the V0 sector to italics. This procedure is used by some authors. We apologize for this inconvenience.
Some sentences have been shortened and corrected.
- The introduction part is not impressive and systematic. In the introduction part, the authors should elaborate on the scientific issues in plant research. The Content of the introduction is effective in essence but very poorly presented, significant improvements are needed in presenting the proper background of the work undertaken
The introduction has been rearranged and improved.
- Summary and Perspective must be replaced with a conclusion and future prospects.
Summary and Perspective have been changed. They have been divided into two parts, the first entitled Coordinating Plant Proton Pump Function (which has been updated as suggested by another reviewer) and the second Conclusions and Future Prospects.
- The conclusion section is very lengthy. The author should emphasize this in a better way.
The Conclusions section has been shortened as previously described.
Please see the attachment
Reviewer 2 Report
To,
The Editor,
IJMS, MDPI,
Manuscript ID: ijms-2196674
Subject: Submission of comments of the manuscript in “IJMS"
Dear Editor IJMS, MDPI,
Thank you very much for the invitation to consider a potential reviewer for the manuscript (ID: ijms-2196674). My comments responses are furnished below as per each reviewer’s comments.
In the reviewed manuscript, the authors focus on two different ATP-driven proton pumps that, despite having many common roles in plant physiology, differ significantly in their structure, regulation, mechanism of action, and location in different membranes. There are some minor and major comments. The author should revise as per my comments and suggestions.
- The abstract does not reflect the whole story, revise it
- The writing style of the paper is very poor. There are lots of grammatical mistakes. Long sentences with noticeable grammatical mistakes are frequently present throughout the manuscript. There are many typos mistakes in this whole manuscript. The author should check the whole manuscript.
- The introduction part is not impressive and systematic. In the introduction part, the authors should elaborate on the scientific issues in plant research. The Content of the introduction is effective in essence but very poorly presented, significant improvements are needed in presenting the proper background of the work undertaken
- Summary and Perspective must be replaced with a conclusion and future prospects
- The conclusion section is very lengthy. The author should emphasize this in a better way.
Author Response
Reviewer 2
Thank you very much for your valuable comments.
- The first sentence of the introduction should be changed to include the two F1F0 ATPase in the chloroplast and mitochondria.
The beginning of the Introduction has been changed as suggested.
- I suggest to add schematically the mechanism of activity of both enzymes.
A diagram illustrating the functioning of proton pumps in plant cells was introduced.
Reviewer 3 Report
1. The first sentence of the introduction should be changed to include the two F1F0 ATPase in the chloroplast and mitochondria.
2. I suggest to add schematically the mechanism of activity of both enzymes.
Author Response

(The authors gave the same response as above.)

Reviewer 4 Report
An interesting, information-rich review of the two main generators of the proton gradient on plant cell membranes is presented: the proton ATPase of the plasmelemma (P-type ATPase) and the proton ATPase of intracellular membranes (V-type ATPase). The review is based on a large array of modern articles.
Two small remarks.
1. The phrase “The P-type ATPases are divided into five subfamilies (P1B, P2A/B, P3A, P4 and P5) [12]» contains an inaccuracy. Firstly, it is better to refer to the fundamental work of Axelsen and Palmgren (1998), in which the classification of P-type ATPases was proposed. Secondly, in accordance with this work, the subfamilies of P-ATPases should be correctly indicated in parentheses (P1A/B, P2A/B/C/D, P3A/B, P4, P5).
Axelsen KB, Palmgren MG. Evolution of Substrate Specificities in the P-Type ATPase Superfamily // J.Mol. Evolution, 1998. 46: 84 – 101.
2. Note to Fig. 2. I suggest the authors to think about how to depict the transmembrane domain “M” in the figure. Alternatively, you can indicate the existence of this domain at least in the caption to Fig.2.
An interesting, information-rich review of the two main generators of the proton gradient on plant cell membranes is presented: the proton ATPase of the plasmelemma (P-type ATPase) and the proton ATPase of intracellular membranes (V-type ATPase). The review is based on a large array of modern articles.
Author Response
Reviewer 3
Two small remarks.
- The phrase “The P-type ATPases are divided into five subfamilies (P1B, P2A/B, P3A, P4 and P5) [12]» contains an inaccuracy. Firstly, it is better to refer to the fundamental work of Axelsen and Palmgren (1998), in which the classification of P-type ATPases was proposed. Secondly, in accordance with this work, the subfamilies of P-ATPases should be correctly indicated in parentheses (P1A/B, P2A/B/C/D, P3A/B, P4, P5).
Axelsen KB, Palmgren MG. Evolution of Substrate Specificities in the P-Type ATPase Superfamily // J.Mol. Evolution, 1998. 46: 84 – 101.
Thank you very much for these suggestions. We have corrected the text and quoted the relevant literature.
- Note to Fig. 2. I suggest the authors to think about how to depict the transmembrane domain “M” in the figure. Alternatively, you can indicate the existence of this domain at least in the caption to Fig.2.
Thank you very much for this tip. Indeed, in Figure 2 we have not marked the membrane domain. Under the picture, we added information about the M domain.
An interesting, information-rich review of the two main generators of the proton gradient on plant cell membranes is presented: the proton ATPase of the plasmelemma (P-type ATPase) and the proton ATPase of intracellular membranes (V-type ATPase). The review is based on a large array of modern articles.
Reviewer 5 Report
The present manuscript has been submitted by Katarzyna Kabala and Malgorzata Janicka, two experts in the field of plant proton pumps. In general, I have missed more information on the linkage between both proton pumps, for instance their cooperative effect on the cytosolic pH might be described in more detail, cell-type specific differences can be considered, too. Such as the role of both proton pumps in guard cells or the function of AHA10 in seeds. In the current version both pumps appear simply side-by-side. Also, I am not convinced, that focusing on ATPase-driven proton pumps reflects the situation in plant cells appropriately. The role of V-PPase and secondary active transporters have effects on proton motif force and pH-homeostasis, too. Last but not least, I would have appreciated more information on the regulation of V-ATPase, there is some information provided on the regulation of the PM-ATPase, but much less for the V-ATPase.
Major points of criticism:
“tightly bound, integral transmembrane protein” sounds weird. I would use “tightly bound” for membrane-associated proteins, which can be tightly or loosely bound, but not for membrane-integral proteins.
Figure 1 would benefit from modelling the pdb-file 5KSD, refer more often to figure 1, use it as support for explaining the structural basis for proton transport (2.2)
2.1 show phylogenetic tree and gene structures as figure
In line 149 you state that “Asp648 is believed to facilitate hydrogen bond formation”, in line 159 “results from hydrogen bond interactions with Asn106”. Please revise.
Line 181 “…A domain catalyzed decay of the phosphoryl group located in the P-domain” Hard to get, revise.
Line 190: “Pump activation results from the removal of the C-terminal R-domain” Removal or movement? Removal would require proteolytic activity, haven’t heard of it before.
2.3 Regulation by phosphorylation and 14-3-3 proteins: provide informations on involved kinases and the respective 14-3-3 family members/subgroups, role of blue light receptors?
Lines 206-221: interesting aspect, provide more information on the function/requirement of modifying PM-ATPases in the transition to land plants.
Line 243: ”of the penultimate threonine and 14-3-3 protein building”, building=binding?
2.4 oligomerization: most data has been published before 2007, there has been much (technical) progress in obtaining structural information on proteins since then, check for new references and improved models of higher resolution.
Line 277: "PMA1p is the primary protein component of the membrane compartment of Pma1 (MCP)", sentence is almost identical to the abstract of Zhao et al. Define MCP as micro domain.
3.1.1 V1 subcomplex: The review cites mostly reviews in this section, this should either be mentioned “reviewed by” or original work should be cited.
Line 459: Tan et al. (2022) used “crude” membrane extracts for isolation of V-ATPases, which contained all endomembranes and not just vacuoles.
3.1.3 I missed more information on the relevance of the interactions with other proteins
Line 520 – 522: that has been shown for the yeast V-ATPase, this should be stated.
Proton transport involves twisting of VHA-c helices
Section on inhibition by chemical compounds (568 – 580) appears to be out of place, better would be one about regulation of V-ATPases.
Table 1 should contain the same organisms mentioned in lines 591 – 593, especially Glycine max as one extreme.
Lines 649 – 652 is this your assumption? Explain more!
Minor points:
Line 14: ATPeses
Line 189: activated Pump activation
Line 307 viewed by 54 (reviewed by)
Line 343: @-subunit of F-ATPases (typo and should be F-ATPsynthases)
Line 647: cooper stress (copper)
Author Response
Reviewer 4
The present manuscript has been submitted by Katarzyna Kabala and Malgorzata Janicka, two experts in the field of plant proton pumps. In general, I have missed more information on the linkage between both proton pumps, for instance their cooperative effect on the cytosolic pH might be described in more detail, cell-type specific differences can be considered, too. Such as the role of both proton pumps in guard cells or the function of AHA10 in seeds. In the current version both pumps appear simply side-by-side. Also, I am not convinced, that focusing on ATPase-driven proton pumps reflects the situation in plant cells appropriately. The role of V-PPase and secondary active transporters have effects on proton motif force and pH-homeostasis, too. Last but not least, I would have appreciated more information on the regulation of V-ATPase, there is some information provided on the regulation of the PM-ATPase, but much less for the V-ATPase.
Thank you very much for your insightful review and all your valuable comments. According to the special issue for which we prepared the manuscript (Structural Biology of Membrane Proteins), our attention focused on the structure of both proteins and the functional aspects related to this structure. It was not our intention to describe the numerous functions of both proteins and their detailed regulation. Of course, these are very important aspects of the functioning of plant H+-ATPases. Recently, the cooperative effect of both proton pumps on the cytosolic pH has been described in detail by Cosse and Seidel (Front. Plant Sci. 12:672873), while their essential functions in plant growth and stress reactions by Li et al. (Annu. Rev. Plant Biol. 2022, 73:495–521). The purpose of our review was to compare the structure of both proteins and the mechanism of their action. We intended to show significant differences in this respect. Also for this reason, we paid attention to their interaction in selected processes only in the summary. We did not want to extend the content of the manuscript too much or repeat the scope of other reviews. Thus, we mentioned some issues without developing them. We understand, therefore, that the reviewer is under the impression that both pumps appear side by side in the manuscript. We did not point out common characteristics, but on the contrary, we focused on the differences
Additional information has been introduced regarding the coordinated action of proton pumps in the regulation of cytosolic pH, stomatal movements and tonoplast P3A-ATPases. In our opinion, expanding the topic with other transporters, including V-PPase, is not directly related to the subject of the review.
The phosphorylation and binding of the 14-3-3 protein by PM ATPase has been discussed to describe the functioning of this pump as an oligomeric complex. Other mechanisms responsible for regulating this protein are not shown. V-ATPase regulation is much more complex. This is due to its multi-subunit structure, within which each of the subunits may be subjected to different regulatory mechanisms. We have introduced a short paragraph on the regulation of V-ATPase in the subsection on the interaction of VHA subunits with other proteins.
Major points of criticism:
“tightly bound, integral transmembrane protein” sounds weird. I would use “tightly bound” for membrane-associated proteins, which can be tightly or loosely bound, but not for membrane-integral proteins.
It has been corrected.
Figure 1 would benefit from modelling the pdb-file 5KSD, refer more often to figure 1, use it as support for explaining the structural basis for proton transport (2.2)
When creating the model presented in Fig. 1, i.e. the schematic structure and domain organization of P-type ATPases, we based, among others, on the model indicated by the reviewer and included in the work of Focht et al. 2017 [30].
Fig. 1 was referred in several places as suggested.
2.1 show phylogenetic tree and gene structures as figure
Based on the phylogenetic tree presented in the work of Wdowikowska and Kłobus 2016 (co-workers from our department), we described in detail which isoforms of the PM H+-ATPase belong to individual subfamilies. Since the special issue is devoted to protein structure, we deliberately did not want to focus much on genes. Given the above, since the phylogenetic tree is available in an earlier publication (Wdowikowska and Kłobus 2016), we have therefore completed the work with a detailed description of which isoforms belong to the appropriate subfamilies.
In line 149 you state that “Asp648 is believed to facilitate hydrogen bond formation”, in line 159 “results from hydrogen bond interactions with Asn106”. Please revise.
It has been corrected.
Line 181 “…A domain catalyzed decay of the phosphoryl group located in the P-domain” Hard to get, revise.
It has been corrected.
Line 190: “Pump activation results from the removal of the C-terminal R-domain” Removal or movement? Removal would require proteolytic activity, haven’t heard of it before.
It has been corrected.
2.3 Regulation by phosphorylation and 14-3-3 proteins: provide informations on involved kinases and the respective 14-3-3 family members/subgroups, role of blue light receptors?
The information has been added to the manuscript.
Lines 206-221: interesting aspect, provide more information on the function/requirement of modifying PM-ATPases in the transition to land plants.
Using AHA2 mutants with several truncated C-terminal variants, the importance of the role of the C-terminal R domain in land plants has been demonstrated [Steger et al., 2023]. The change in the C-terminus contributed to the opening of the stomatal pores, which was disadvantageous as it allowed excessive water loss and entry of pests. The mutated plants were thus more susceptible to biotic and abiotic stresses, such as pathogen invasion and water loss. Based on this study authors (Anette Steger et al. 2022) believe that the R domain is crucial for the fitness of land plants. The information has been added to the manuscript.
Line 243: ”of the penultimate threonine and 14-3-3 protein building”, building=binding?
It has been corrected.
2.4 oligomerization: most data has been published before 2007, there has been much (technical) progress in obtaining structural information on proteins since then, check for new references and improved models of higher resolution.
When describing oligomerization, we relied on older works where this process was described for the first time (publications from 1995-2007), but also on the latest works from 2018-2021.
- Nguyen, T.T.; Sabat, G.; Sussman, M.R. In vivo cross-linking supports a head-to-tail mechanism for regulation of the plant plasma membrane P-type H+-ATPase. J. Biol. Chem. 2018, 293, 17095–17106.
- Zhao, P.; Zhao, Ch.; Chen, D.; Yun, C.; Li, H.; Bai, L. Structure and activation mechanism of the hexameric plasma membrane H+-ATPase. Nature Commun. 2021, 12, 6439.
- Heit, S.; Geurts, M.M.G.; Murphy, B.J.; Corey, R.A.; Mills, D.J.; Kühlbrandt, W., et al. Structure of the hexameric fungal plasma membrane proton pump in its autoinhibited state. Sci. Adv. 2021, 7(46), eabj5255.
- Lapshin, N.K.; Piotrovskii, M.S.; Trofimowa M.S. Sterol Extraction from Isolated Plant Plasma Membrane Vesicles Affects H+-ATPase Activity and H+-Transport. Biomolecules 2021, 11, 1891.
Line 277: "PMA1p is the primary protein component of the membrane compartment of Pma1 (MCP)", sentence is almost identical to the abstract of Zhao et al. Define MCP as micro domain.
This sentence did not add any significant information to the paper and was removed from the manuscript.
3.1.1 V1 subcomplex: The review cites mostly reviews in this section, this should either be mentioned “reviewed by” or original work should be cited.
We agree with the reviewer's suggestion, appropriate changes have been made.
Line 459: Tan et al. (2022) used “crude” membrane extracts for isolation of V-ATPases, which contained all endomembranes and not just vacuoles.
It has been corrected.
3.1.3 I missed more information on the relevance of the interactions with other proteins
Information on the interaction of VHA subunits with other proteins has been supplemented.
Line 520 – 522: that has been shown for the yeast V-ATPase, this should be stated.
It has been corrected.
Proton transport involves twisting of VHA-c helices
This information was included in the paragraph describing conformational changes of V-ATPase subunits during rotation.
Section on inhibition by chemical compounds (568 – 580) appears to be out of place, better would be one about regulation of V-ATPases.
The paragraph concerning inhibitors, in the opinion of the authors, is related to the main subject of the work, i.e. the structure of proton pumps. Its purpose was to draw attention to the important points/residues, the blocking of which is associated with the inhibition of the c-ring rotation. However, at the suggestion of the reviewer, we have removed this paragraph.
Table 1 should contain the same organisms mentioned in lines 591 – 593, especially Glycine max as one extreme.
The table was updated according to the suggestion and a new version was introduced.
Lines 649 – 652 is this your assumption? Explain more!
This is not our assumption. The complex structure of V-ATPase and numerous isoforms of individual subunits give the possibility of some structural flexibility and the formation of various complexes functioning in different organs, developmental stages and organelles (Seidel 2022). For many years, it has been believed that V-ATPase acts as an enzyme that responds to changing physiological and environmental conditions. Expression of specific isoforms may be related not only to plant development but also to plant adaptation. In turn, changes at the expression level of individual subunits may be responsible for the modulation of the enzyme structure. For this reason, V-ATPase has been called an "eco-enzyme" (Ratajczak 2000). However, nowadays this term is less used.
We have added appropriate citations in the text.
Minor points:
Line 14: ATPeses
It has been corrected.
Line 189: activated Pump activation
It has been corrected.
Line 307 viewed by 54 (reviewed by)
It has been corrected.
Line 343: @-subunit of F-ATPases (typo and should be F-ATPsynthases)
We apologize for this typo error. It was created when transferring the file to the template. It has been corrected. The phrase "F-ATPase" has been changed to ATP synthase, also in other parts of the text.
Line 647: cooper stress (copper)
It has been corrected.
Round 2
Reviewer 5 Report
Thank you for explaing the intention of your manuscript, now I understand the focus of your review. Also, you are right that it makes sense to have a different scope than other reviews to avoid redundancy. From my point of view, you have sucessfully improved the manuscript so that it is ready for publication.